# Characterization of a Novel Splicing Variant in Acylglycerol Kinase (AGK) Associated with Fatal Sengers Syndrome

**DOI:** 10.3390/ijms222413484

**Published:** 2021-12-15

**Authors:** Sofia Barbosa-Gouveia, Maria E. Vázquez-Mosquera, Emiliano Gonzalez-Vioque, Álvaro Hermida-Ameijeiras, Laura L. Valverde, Judith Armstrong-Moron, Maria del Carmen Fons-Estupiña, Liesbeth T. Wintjes, Antonia Kappen, Richard J. Rodenburg, Maria L. Couce

**Affiliations:** 1Unit of Diagnosis and Treatment of Congenital Metabolic Diseases, Department of Paediatrics, Santiago de Compostela University Clinical Hospital, 15706 Santiago de Compostela, Spain; maria.eugenia.vazquez.mosquera@sergas.es (M.E.V.-M.); alvaro.hermida@usc.es (Á.H.-A.); laura261lv@gmail.com (L.L.V.); 2Rare Diseases Networking Biomedical Research Centre (CIBERER), IDIS-Health Research Institute of Santiago de Compostela, Santiago de Compostela University Clinical Hospital, European Reference Network for Hereditary Metabolic Disorders (MetabERN), 15706 Santiago de Compostela, Spain; 3Clinical Biochemistry Service, Puerta de Hierro-Majadahonda University Hospital, 28222 Majadahonda, Spain; egvioque@gmail.com; 4Clinical Genetics, Molecular and Genetic Medicine Section, Hospital Sant Joan de Déu and Institut de Recerca Sant Joan de Déu, 08950 Esplugues de Llobregat, Spain; jarmstrong@sjdhospitalbarcelona.org; 5CIBER-ER (Biomedical Network Research Center for Rare Diseases), Instituto de Salud Carlos III (ISCIII), 28029 Madrid, Spain; 6Pediatric Neurology Department, Institut de Recerca Sant Joan de Déu, Sant Joan de Déu University Hospital, 08950 Barcelona, Spain; cfons@sjdhospitalbarcelona.org; 7Translational Metabolic Laboratory, Department of Laboratory Medicine, Radboud University Medical Centre, 6525 GA Nijmegen, The Netherlands; liesbeth.Wintjes@radboudumc.nl (L.T.W.); antonia.vanheck-kappen@radboudumc.nl (A.K.); 8Department of Pediatrics, Radboud Centre for Mitochondrial Medicine, Radboud University Medical Centre, 6525 GA Nijmegen, The Netherlands; richard.rodenburg@radboudumc.nl

**Keywords:** acylglycerol kinase, mitochondrial dysfunction, mitochondrial ATP generation, oxidative phosphorylation machinery, Sengers syndrome

## Abstract

Mitochondrial functional integrity depends on protein and lipid homeostasis in the mitochondrial membranes and disturbances in their accumulation can cause disease. *AGK*, a mitochondrial acylglycerol kinase, is not only involved in lipid signaling but is also a component of the TIM22 complex in the inner mitochondrial membrane, which mediates the import of a subset of membrane proteins. *AGK* mutations can alter both phospholipid metabolism and mitochondrial protein biogenesis, contributing to the pathogenesis of Sengers syndrome. We describe the case of an infant carrying a novel homozygous *AGK* variant, c.518+1G>A, who was born with congenital cataracts, pielic ectasia, critical congenital dilated myocardiopathy, and hyperlactacidemia and died 20 h after birth. Using the patient’s DNA, we performed targeted sequencing of 314 nuclear genes encoding respiratory chain complex subunits and proteins implicated in mitochondrial oxidative phosphorylation (OXPHOS). A decrease of 96-bp in the length of the *AGK* cDNA sequence was detected. Decreases in the oxygen consumption rate (OCR) and the OCR:ECAR (extracellular acidification rate) ratio in the patient’s fibroblasts indicated reduced electron flow through the respiratory chain, and spectrophotometry revealed decreased activity of OXPHOS complexes I and V. We demonstrate a clear defect in mitochondrial function in the patient’s fibroblasts and describe the possible molecular mechanism underlying the pathogenicity of this novel *AGK* variant. Experimental validation using in vitro analysis allowed an accurate characterization of the disease-causing variant.

## 1. Introduction

Mitochondria are responsible for several vital functions in mammalian cells, including adenosine triphosphate (ATP) production, steroid and stress hormones synthesis, cellular metabolism, and the induction of apoptosis [1,2,3]. Moreover, spatially and functionally regulated specific microdomains known as the mitochondria-associated membranes (MAMs) between the endoplasmic reticulum (ER) and mitochondria are hot spots of Ca^2+^ transfer, highlighting the mitochondria vital role in cellular Ca^2+^ homeostasis [4].

Mitochondrial diseases can arise due to defects in nuclear or mitochondrial DNA (nDNA and mtDNA, respectively) genes that encode proteins required for normal mitochondrial structure and/or function [5]. Acylglycerol kinase (AGK) is a mitochondrial membrane protein involved not only in lipid and glycerolipid metabolism but also in mitochondrial protein transport, glycolysis, and thrombocytopoiesis [6,7,8]. AGK can act as a lipid kinase to catalyze the phosphorylation of diacylglycerol (DAG) and monoacylglycerol (MAG) to phosphatidic acid (PA) and lysophosphatidic acid (LPA), respectively. Both PA and LPA can act as signaling molecules and participate in phospholipid synthesis [9,10], which is essential to maintain mitochondrial structure and function. Moreover, AGK scavenges mitochondrial DAG, which induces reactive oxygen species (ROS) signaling via a pathway mediated by protein kinase D1 [9,11].

Independently of its lipid kinase activity and its role in lipid metabolism stability, AGK acts as a subunit of the mitochondrial translocase of the inner membrane 22 (TIM22) complex [12,13], which is responsible for the translocation of transmembrane proteins from the cytoplasm into the mitochondrial interior [14] and their insertion into the mitochondrial inner membrane via the formation of a twin-pore translocase that harnesses membrane potential as the external driving force [14,15]. The TIM22 complex, which is required to import a subset of metabolite carriers into mitochondria, including ANT1/SLC25A4 and SLC25A24, consists of multiple subunits: Tim22, the core pore-forming subunit; the intermembrane space chaperones Tim9, Tim10, and Tim10b; and AGK and Tim29, which are specific subunits of the complex [14].

Sengers syndrome (OMIM #212350) is a rare autosomal recessive disorder caused by mutations in the *AGK* gene and characterized by hypertrophic cardiomyopathy, congenital cataracts, and mitochondrial myopathy, including muscle weakness and lactic acidosis after exercise [16]. Two clinical forms have been identified: a severe neonatal form that can cause infantile death due to heart failure consequent to hypertrophic cardiomyopathy, and a benign form with a better prognosis in which surviving patients achieve normal developmental milestones [9,17]. The pathological mechanism underlying Sengers syndrome remains unclear, although it is thought to involve AGK’s role in lipid metabolism.

In this study, we sought to characterize the possible molecular mechanism underlying the pathogenicity of a novel homozygous *AGK* variant detected in a patient with suspected Sengers syndrome. We discuss the importance of experimental validation using in vitro analysis for the accurate characterization of disease-causing variants.

## 2. Results

### 2.1. Molecular Genetics and In Silico Analysis of the AGK Variant

NGS analysis enabled the identification of a homozygous variant, c.518+1G>A, in the *AGK* gene (NM_018238.4) in this patient. This splicing variant is located in intron 8, and segregation studies performed by Sanger sequencing confirmed that it was inherited from both parents (Figure 1A). The *AGK* variant was analyzed in silico to determine the degree of evolutionary conservation, predicted pathogenicity, functional consequences, and minor allele frequency (MAF) within the population (Appendix A). In addition, the Human Splicing Finder (HSF) [18] predicted that the splicing variant c.518+1G>A, located in intron 8, disrupts the wild-type splice donor site, likely affecting splicing (Figure 1B). Variants in the canonical acceptor and donor sites have been previously reported to strongly affect conserved sequences [19]. The 5′ splice site (donor site) and 3′ splice site (acceptor site) sequences are recognized by the elements of the spliceosome. Consequently, any variant located in these canonical sequences can alter the interactions between pre-mRNA and proteins involved in intron removal. Variants at the canonical splice sequences usually lead to single exon skipping [20]. The *AGK* gene encodes a 422-amino acid protein. Sequencing of our patient’s cDNA revealed a decrease in sequence length of 96 bp compared with control cDNA (Figure 1C), an effect likely due to exon 9 (96 bp) skipping during the splicing process.

### 2.2. AGK Protein Modeling

The AGK protein consists of a typical two-domain fold (DGK domain 1 and DGK domain 2) that mediates phosphorylation of monoacylglycerols or diacylglycerols and has an N-terminal α1 helix that anchors to the membrane and a C-terminal key region with an additional membrane anchor helix loop (Figure 2) [6,21].

### 2.3. Mitochondrial Respiration and OXPHOS Activity

Measurement of the oxygen consumption rate (OCR) and extracellular acidification rate (ECAR) revealed decreases in both OCR and the OCR:ECAR ratio in the patient’s fibroblasts, indicating reduced electron flow through the respiratory chain (Figure 3A,B). These findings reflect a general deficiency in the overall function of all mitochondrial respiratory chain complexes in the patient’s fibroblasts. 

Spectrophotometric analysis of respiratory chain enzyme activity revealed deficiencies in complexes I and V in the patient’s fibroblasts (Table 1).

## 3. Discussion

We present the results of molecular and functional studies supporting the pathogenicity of the homozygous *AGK* variant c.518+1G>A, which is reported for the first time. Through molecular characterization using the patient’s own fibroblasts, we have demonstrated mitochondrial bioenergetic dysfunction, which leads to defective mitochondrial respiration with a significant decrease in the OCR:ECAR ratio, and defective OXPHOS activity due to decreased activity of complexes I and V. Patients with Sengers syndrome, an autosomal recessive disorder, usually present with congenital cataracts, lactic acidosis, skeletal myopathy, and hypertrophic cardiomyopathy [16]. Interestingly, most *AGK* pathogenic variants are nonsense mutations or occur at splice sites, as in our patient, affecting mRNA splicing. The phenotype–genotype correlation reported in previous studies suggests that homozygous *AGK* nonsense variants result in a severe form of Sengers syndrome, while patients harboring splice site variants can survive the first decade of life [17,22]. Interestingly, the patient described here presented a severe form of Sengers syndrome and harbored a splice site mutation. The severity of Sengers syndrome can vary widely depending on the nature and location of the variant (Appendix A) [9,17]. For instance, one of 2 siblings (P4 and P5) carrying the homozygous variant c.1131+5G>A, which leads to a splicing defect, died at age 12, while the other was still alive at age 10 [23]. Patient P3 with the same homozygous variant had a later onset of the disease and was still alive at the age of 41 [24]. On the other hand, isolated congenital cataracts was the only clinical sign observed in three patients (P13, P14 and P15) carrying the homozygous p.Ala142Thrfs*4 mutation, which resulted in complete deletion of exon 8 and premature truncation of the protein [22]. The onset of the disease at birth is likely associated with protein-truncating variants such as nonsense, frameshift, splice acceptor, and splice donor mutations. Although, frameshift and splicing defects were identified in patients with a later onset (P3, P4 and P33). Moreover, it seems homozygous variants frequently lead to a fatal prognosis of Sengers syndrome, while patients with compound heterozygous variants might live longer.

AGK was initially identified as a mitochondrial lipid kinase involved in the synthesis of PA, which inhibits mitochondrial division and stimulates mitochondrial outer membrane fusion [25]. Cardiolipin (CL), a mitochondria-specific phospholipid, is synthesized from PA in the inner mitochondrial membrane. Here it plays a crucial role in the maintenance of large protein complexes, including electron transport chain complexes and the protein import machinery of the inner membrane [25]. Recent analyses have revealed specific interaction sites for CL and shown that CL is required for full activity of complexes I and V [26,27]. In fact, complex V, which uses energy stored in the proton gradient across the inner membrane for the synthesis of ATP, binds to CL, for which a role in proton translocation at complex V has been proposed. The mitochondria play an important role in organs with high energy demands, such as the heart. CL is a key player in several cellular processes and pathways that are crucial for heart function, including autophagy/mitophagy, mitochondrial function, and mitochondrial protein import [28]. In addition, the link between CL and cardiovascular diseases could be explained by the physiological role of CL in cardioprotective signaling pathways and the mitogen-activated protein kinase (MAPK) pathway. While several studies have demonstrated decreased CL levels in heart failure in both humans [29,30] and rats [31], these decreases may contribute to a progressive reduction in mitochondrial activity, with consequent severe effects on energy metabolism, as occurs in cardiac disease, resulting in cardiomyopathy, heart attack, cardiac valvular disease, atrial fibrillation, and high blood pressure. Indeed, our patient exhibited reduced complex I and V activity, and defects in CL metabolism, as reflected by congenital dilated myocardiopathy during the neonatal period.

AGK plays a kinase-independent role as a subunit of the TIM22 complex, a mitochondrial translocase involved in complex stability and import of mitochondrial carrier proteins into the mitochondrial inner membrane [32,33]. The human TIM22 complex consists of TIM22, the core pore-forming unit; TIM29; the intermembrane space chaperones TIM9, TIM10, and TIM10B; and the lipid kinase AGK [14]. Two main classes of the TIM22 complex are described: SLC25, the largest protein family of mitochondrial solute transporters, which act as inner membrane metabolite transporters; and TIM proteins, which possess transmembrane domains and serve as components of the inner membrane protein translocase complexes [14]. Since the TIM22 complex is a multisubunit molecular machine specialized for the translocation of proteins with internal targeting signals into the inner mitochondrial membrane, defects in *AGK* can result in defective mitochondrial protein import and the possibility of developing Sengers syndrome due to depletion of TIM22 complex substrates. In fact, a recent study of patients with *AGK* mutations reported decreased levels of TIM22 complex and metabolite carrier import into mitochondria [13].

Two Sengers syndrome phenotypes are described: a severe neonatal form leading to infantile death; and a mild chronic form in which affected patients survive into the fourth decade of life [34]. Evidence of a genotype–phenotype correlation in Sengers syndrome remains unclear, as the precise molecular function of *AGK* is yet to be elucidated. On the one hand, its potential role in CL metabolism is consistent with the varied bioenergetic impairment, structurally abnormal mitochondria, and lipid and glycogen deposits in patient-derived skeletal and cardiac muscle. Because the heart is one of the organs with the greatest energy demands, the OXPHOS system plays an essential role in cardiac metabolism. On the other hand, the Sengers syndrome phenotype overlaps with those of protein biogenesis disorders, such as mutation in the mitochondrial adenine nucleotide translocator (ANT) [35], and of diseases of lipid metabolism, such as Barth syndrome [36]. Moreover, genetic modifiers can significantly affect disease progression and severity [34]. A clearer understanding of the manifold functions of *AGK* is essential to shed further light on the pathogenic mechanisms underlying Sengers syndrome. Moreover, this knowledge will facilitate the design of therapeutic strategies for this disorder, as already demonstrated for gene replacement therapy approaches for similar disorders [37].

## 4. Materials and Methods

### 4.1. Study Design

This study was carried out in collaboration with the University Clinical Hospital of Santiago de Compostela (Spain) and Radboud University Medical Center (The Netherlands). Parents of all patients provided written informed consent to participate in the study and publication of the results. All experimental protocols were approved by the Rabdoud University Medical Center and were performed in accordance with relevant guidelines and regulations.

### 4.2. Clinical Profile

Our patient was a 1-day-old boy with congenital cataracts, pielic ectasia and critical congenital dilated cardiomyopathy who was born after induction of labor at 38 weeks gestation with cardiac volume overload and signs of congestive heart failure. Birth weight was 2750 g (20th percentile), length 50 cm (80th percentile), and head circumference 33 cm (20th percentile). The patient was the first child born to healthy consanguineous parents. At birth, the newborn was clinically stable with APGAR scores of 3 and 8 at 1 and 5 min, respectively, but he was promptly placed on continuous positive airway pressure (CPAP) and diuretics due to metabolic acidosis (pH 7.20; bicarbonate, 12 mmol/L; lactate 9.8 mmol/L).

Hyperlactacidemia worsened and serum lactate levels increased rapidly up to 17 mmol/L during the first 12 h after birth. Further laboratory blood tests revealed a prolonged partial thromboplastin time (PTT) with normal transaminases, bilirubin, and albumin. Plasma amino acid analysis revealed elevation of alanine, methionine, and glutamine, reflecting liver dysfunction and a direct influence of acidosis on the moderate increase in protein synthesis. The results of the bacterial culture of fluids were negative. Imaging studies included a head ultrasound revealing normal ventricles and white matter. An echocardiogram revealed severely depressed left ventricular systolic function (left ventricular ejection fraction, 40%) mainly due to dyskinesia of the posterior wall.

The severity of lactic acidosis continued to increase despite initiation of continuous vasopressor infusion and ventilator support, resulting in deterioration of the patient’s general and hemodynamic status followed by sudden cardiac arrest and death 20 h after birth.

### 4.3. Targeted Next-Generation Sequencing

DNA was isolated from 400 μL of the patient’s blood collected in EDTA following standard procedures and analyzed with targeted next-generation sequencing (NGS) panels for mitochondrial diseases. Targeted analysis was performed on 314 nuclear genes encoding respiratory chain complex subunits and proteins implicated in OXPHOS function (Appendix A). Genetic data were analyzed by NGS technology through a process consisting of enrichment with an in-solution hybridization technology (Sure Select XT; Agilent Technologies, Santa Clara, CA, USA) followed by subsequent sequencing on a NextSeq platform (Illumina Inc., San Diego, CA, USA). A custom SureSelect probe library was designed to capture the exons and exon-intron-boundaries of the targeted genes [38]. Sequence capture, enrichment, and elution were performed in accordance with the manufacturer’s instructions. To ensure reliable clinical interpretation of the detected variants, prioritization criteria were applied to predict their pathogenicity in accordance with AMCG guidelines [39].

### 4.4. Cell Culture

Patient and control fibroblasts were cultured in M199 medium (Gibco, Life Technologies, Grand Island, CA, USA) supplemented with 10% *v*/*v* fetal calf serum (FCS) and 1% *v*/*v* penicillin/streptomycin (Gibco, Life Technologies, Grand Island, CA, USA) at 37 °C and 5% CO_2_. 

### 4.5. Mitochondrial Isolation

Pelleted fibroblasts from the patient and from the control cell line were resuspended in ice cold Tris-HCl (10 mM, pH 7.6). Cells were then disrupted in a Potter-Elvejhem homogenizer at 1800 rpm and sucrose (250 mM) was added to make the samples isotonic. The homogenized cell samples were then centrifuged for 10 min at 600× *g* and the mitochondria pellet was obtained after centrifugation of the supernatant for 10 min at 14,000× *g*.

### 4.6. Respirometry and Measurement of OXPHOS Activity

A Seahorse XFe96 Extracellular Flux analyzer (Seahorse Bioscience, Agilent Technologies, Santa Clara, CA, USA) was used to measure the OCR via a fluorophore sensitive to changes in oxygen concentration. The day before the assay, control and patient fibroblasts were seeded at 10,000 cells per well in cell culture medium (M199 supplemented with 10% FCS and 1% pen/strep) and incubated overnight at 37 °C and 5% CO_2_. On the day of the assay, cell culture medium was replaced with Agilent Seahorse XP Base Medium (Agilent Technologies, CA, USA) containing 10 mM glucose (Sigma-Aldrich, St. Louis, MO, USA), 1 mM sodium pyruvate (Gibco, Life Technologies, CA, USA), and 200 mM L-glutamine (Life Sciences Group, Barnet, UK), and then incubated for 1 h at 37 °C without CO_2_. The Seahorse XF Cell Mito Stress Test uses compounds of respiration that target components of the electron transport chain (ETC) in the mitochondria to reveal key parameters of metabolic function (Appendix A). Baseline cellular OCR was measured 8 times followed by 4 measurement cycles after addition of the following inhibitors: 1 µM oligomycin A (Sigma-Aldrich, MO, USA); 2.0 µM + 4.0 µM carbonyl cyanide 4-(trifluoromethoxy) phenylhydrazone FCCP (Sigma-Aldrich, MO, USA); 0.5 µM rotenone + 0.5 µM antimycin A (Sigma-Aldrich, MO, USA). After OCR measurements, the cell medium was removed and replaced with 0.33% Triton X-100, 10 mM Tris-HCl (pH 7.6). Seahorse plates were stored at −80 °C and thawed afterward. Citrate synthase activity was measured spectrophotometrically at 37 °C using a Tecan Spark spectrophotometer. The assay mix contained 0.3 mM acetyl-CoA, 0.1 mM DTNB, 0.025% Triton X-100, and 10 mM Tris-HCl (pH 8.1). Measurements were based on the absorption at 412 nm of thionitrobenzoic acid (TNB), and citrate synthase activity was calculated based on the rate of dithionitrobenzoic acid (DTNB) conversion in the presence of oxaloacetate. OCR was measured before and after the addition of inhibitors and normalized to citrate synthase activity, as described by Srere et al. [40].

The enzymatic activities of complexes I–V, citrate synthase, and protein were assayed spectrophotometrically as previously described [41]. All assays were performed in duplicate using a Konelab 20XT auto-analyzer (Thermo Fisher Scientific, Waltham, MA, USA).

### 4.7. RNA Extraction, cDNA Synthesis and Sequencing

For transcript sequencing, total RNA was extracted from fibroblasts using the GeneJet RNA Purification Mini Kit with quality control and quantification for downstream applications. To study the transcript sequence, the complementary cDNA fragments were synthesized using reverse transcriptase PCR with NZY First-Strand cDNA Synthesis Kit. This sample was quantified and used in a normal polymerase chain reaction (PCR) protocol to check base-pair differences between the control and patient using TapeStation 4200 High Sensitivity D1000 kit. Primers used: 6F—GGGACAAGCCAAGAAACTCCT; 16R—AGCAGTTTCACCTCCACAGG.

### 4.8. Protein Modeling

The *AGK* gene encodes a 47,137-Da protein composed of 422 amino acids. The AGK (Uniprot Q53H12; ID: AGK_HUMAN) protein FASTA sequence was used to build a protein model with SWISS-MODEL. The SWISS-MODEL template library (SMTL version 2021-10-06, PDB release 2021-10-01) was searched with BLAST (Basic Local Alignment Search Tool) [42] and HHBlits (hidden Markov model (HMMs)-based lightning-fast iterative sequence search) [43] for evolutionary-related structures matching the target sequence. Using this homology model, we investigated how the homozygous splicing variant detected in our patient could alter the protein structure, potentially resulting in conformational changes with a significant impact on function (most likely loss of function).

## 5. Conclusions

In summary, we show that the homozygous variant described here for the first time exerts a molecular impact on the integrity of the mitochondrial respiration system in the patient’s fibroblasts. This alteration may cause an imbalance in mitochondrial metabolites and lead to secondary dysfunction of OXPHOS. However, it remains to be determined whether the pathogenic mechanism underlying Sengers syndrome is due predominantly to loss of AGK lipid kinase activity or loss of AGK protein import activity, or indeed a combination of both.

## Figures and Tables

**Figure 1 ijms-22-13484-f001:**
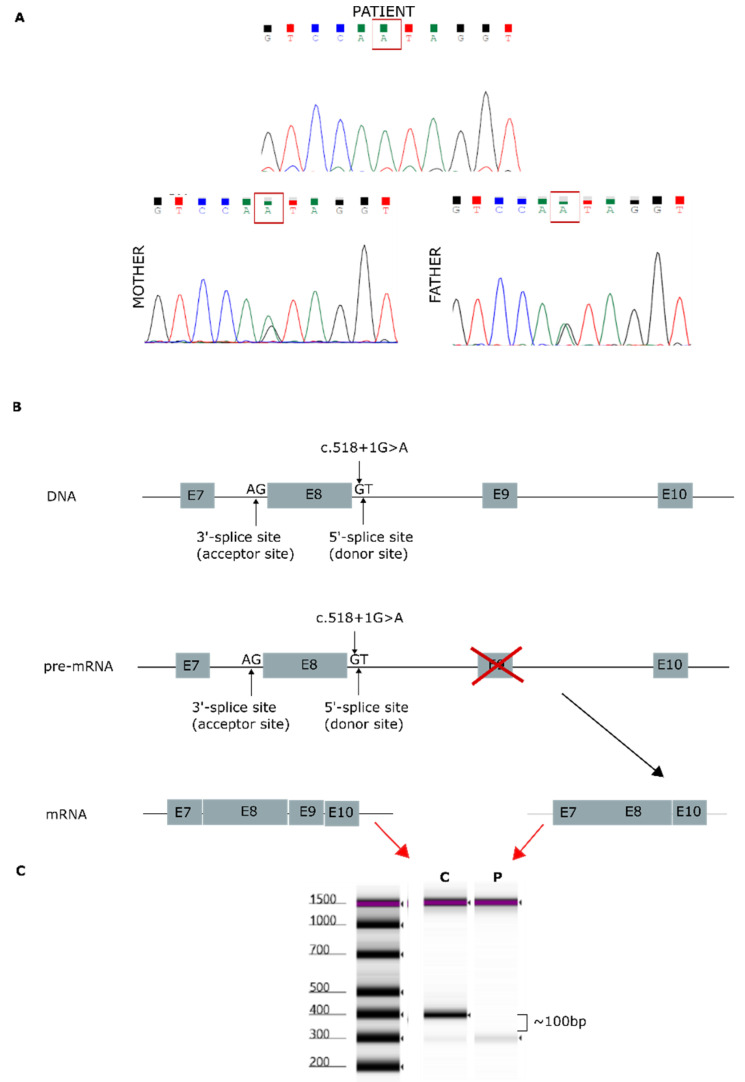
Genetic analysis findings. (**A**) Genomic DNA sequence chromatograms showing Sanger sequencing results. Both parents carried the splicing variant c.518+1G>A, located in intron 8. (**B**) *AGK* gene: predicted effect of exon 9 skipping during mature mRNA splicing. (**C**) cDNA sequencing of patient [P] and control [C], showing a 96-bp decrease in the length of the *AGK* cDNA sequence.

**Figure 2 ijms-22-13484-f002:**
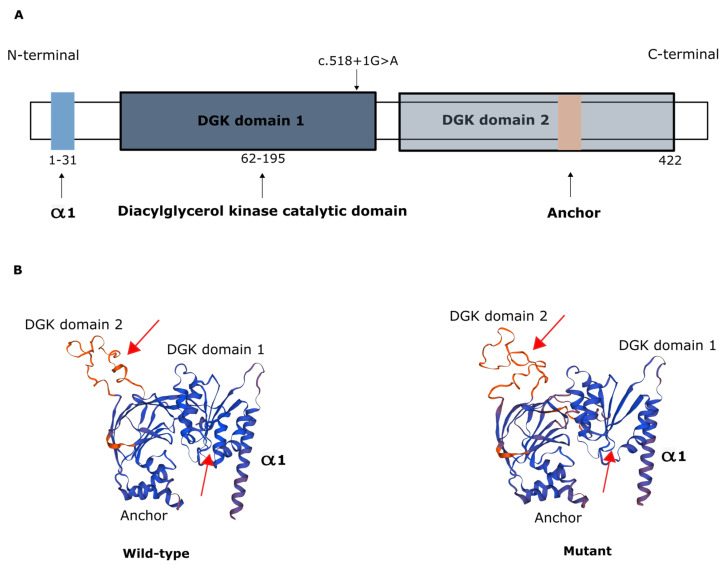
Representation of AGK protein domains. (**A**) Domain structure of AGK, which contains a two-domain fold (DGK domains 1 and 2) that mediates phosphorylation of monoacylglycerols or diacylglycerols, an N-terminal α1 helix (light blue) that anchors to the membrane, and a C-terminal region with an additional membrane anchor (light brown) helix loop. Schematic depicts the c.518+1G>A variant in the catalytic domain. (**B**) Protein modeling of wild-type AGK and of the AGK variant c.518+1G>A, showing predicted changes in spatial protein structure (red arrows).

**Figure 3 ijms-22-13484-f003:**
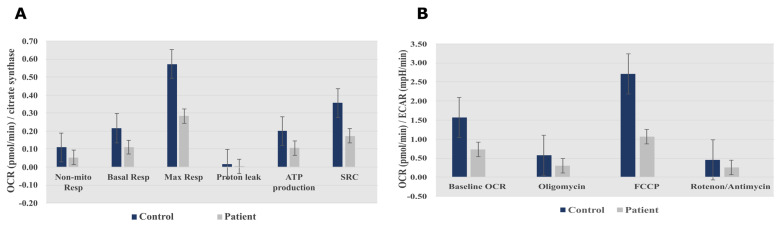
Seahorse XF96 respirometry analysis. (**A**) Oxygen consumption rate measured before and after the addition of inhibitors. These modulators are electron transport chain inhibitors (oligomycin, FCCP, and a mixture of rotenone and antimycin A), which are serially injected to measure ATP production, Max Resp, Non-Mito Resp, proton leak, SRC, and Basal Resp. (**B**) The basal energy metabolism of each cell line was assessed by determining OCR:ECAR ratios following sequential injection of the inhibitors. Abbreviations: Basal Resp, basal respiration; ECAR, extracellular acidification rate; FCCP, carbonyl cyanide-p-trifluoromethoxyphenylhydrazone; Max Resp, maximal respiration; Non-Mito Resp, non-mitochondrial respiration; OCR, oxygen consumption rate; SRC, spare respiratory capacity. All data were obtained from two independent experiments, expressed as mean value ± deviation standard.

**Table 1 ijms-22-13484-t001:** Measurement of enzyme activities for the different OXPHOS complexes in patient and control fibroblasts (reference range determined for healthy population).

	Patient-Activity (mU/U CS)	Control-Activity (mU/U CS)	Reference Range
**Complex I**	**142**	**320**	**163–599**
Complex II	367	582	335–888
Complex III	592	628	570–1383
Complex IV	556	527	288–954
**Complex V**	**103**	**712**	**193–819**

Bold font indicates complex for which reduced activity was observed. Abbreviations: CS, citrate synthase.

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
