# Peer review of "Characterization of a Novel Splicing Variant in Acylglycerol Kinase (AGK) Associated with Fatal Sengers Syndrome"

_ijms, 2021, doi:10.3390/ijms222413484_

Round 1

Reviewer 1 Report

                                                                 COMMENTS TO THE AUTHORS

 IJMS-1495764: “Characterization of a novel splicing variant in AGK associated with fatal Sengers syndrome

Dear Authors,

Please find enclosed the comments for the above-mentioned manuscript.

A SUMMARY OF THE CONTENT

The authors stated that the study intends to describe the case of an infant carrying a novel homozygous Acylglycerol kinase (AGK) variant, c.518+1G>A, who was born with congenital cataracts, pielic ectasia, critical congenital dilated myocardiopathy, and hyperlactacidemia and died 20 hours after birth. The targeted sequencing of nuclear genes encoding respiratory chain complex subunits and proteins implicated in mitochondrial oxidative phosphorylation (OXPHOS) was performed. The results showed decreased the oxygen consumption rate and the extracellular acidification rate ratio in the patient’s fibroblasts as well as decreased activity of OXPHOS complexes I and V. The authors concluded that their results demonstrate a clear defect in mitochondrial function in the patient’s fibroblasts, and describe the molecular mechanism underlying the pathogenicity of the novel AGK variant, pointing that the experimental validation using in vitro analysis allowed an accurate characterization of the disease-causing variant.

THE OVERALL OPINION OF THE MANUSCRIPT

The manuscript is within the scope of the journal and describes the important topic. The text is comprehensive but very easy to follow. The topic is of interest to the readers of IJMS.

(1) TITLE

Please consider to include “acylglycerol kinase” before (AGK) in the title.

(2) ABSTRACT

2.1. Please focus on the aim, methodology and results, rather than introduction (8 rows out of 17.5 describe background).

2.2. Please replace “the molecular mechanism” with “possible molecular mechanism” since the precise mechanism was not described.

(3) INTRODUCTION

3.1. Please include more precise information in the sentence:”Mitochondria are responsible for several vital functions in mammalian cells, including ATP production, cellular metabolism, and the induction of apoptosis” (page 2, lines 51 and 52). Namely, besides mentioned, the mitochondria are the store of Ca2+, important for Ca2+ shuttle with ER, steroid hormones synthesis, stress hormones synthesis, stress response etc.

3.2. Please replace “the molecular mechanism” with “possible molecular mechanism” since the precise mechanism was not described. 

(4) METHODS

4.1. Please provide statement and number of the decision of the Ethical committee.

4.2. Please provide intra- and inter- assay coefficients for all assays.

(5) RESULTS

5.1. Please be more clear and precise in the description of the results.

5.2. Please replace “the molecular mechanism” with “possible molecular mechanism” since the precise mechanism was not described. 

(6) DISCUSSION

6.1. Please replace “the molecular mechanism” with “possible molecular mechanism” since the precise mechanism was not described. 

6.2. Please discuss the association/correlation between different mutations and the manifestation of the symptoms in term of age and/or sex, as well as same/similar mutations in relation to the symptoms and age and sex.  

(7) GENERAL

Please use official abbreviations.

Good luck and all the best :)

Author Response

THE OVERALL OPINION OF THE MANUSCRIPT

The manuscript is within the scope of the journal and describes the important topic. The text is comprehensive but very easy to follow. The topic is of interest to the readers of IJMS.

ANSWER: We would like to thank the reviewer for a careful and thorough reading of our manuscript and the very positive and kind comments.

  1. TITLE. Please consider to include “acylglycerol kinase” before (AGK) in the title.

ANSWER: Thank you very much for the comment. Title updated to "Characterization of a novel splicing variant in acylglycerol kinase (AGK) associated with fatal Sengers syndrome"

  1. ABSTRACT

2.1. Please focus on the aim, methodology and results, rather than introduction (8 rows out of 17.5 describe background).

ANSWER: Following the reviewer suggestion, we have updated the following:

We shortened the introduction:

“Mitochondrial functional integrity depends on protein and lipid homeostasis in the mitochondrial membranes and disturbances in their accumulation can cause disease. AGK, a mitochondrial acylglycerol kinase, is not only involved in lipid signaling, but also a component of the TIM22 complex in the inner mitochondrial membrane, which mediates the import of a subset of membrane proteins. AGK mutations can alter both phospholipid metabolism and mitochondrial protein biogenesis, contributing to the pathogenesis of Sengers syndrome”.  

And we expanded the results section: A decrease of 96-bp in the length of the AGK cDNA sequence was detected.

2.2. Please replace “the molecular mechanism” with “possible molecular mechanism” since the precise mechanism was not described.

ANSWER: Corrected to “possible molecular mechanism”

  1. INTRODUCTION

3.1. Please include more precise information in the sentence: “Mitochondria are responsible for several vital functions in mammalian cells, including ATP production, cellular metabolism, and the induction of apoptosis” (page 2, lines 51 and 52). Namely, besides mentioned, the mitochondria are the store of Ca2+, important for Ca2+ shuttle with ER, steroid hormones synthesis, stress hormones synthesis, stress response etc.

ANSWER: Following reviewer suggestion, we have changed the sentence: “Mitochondria are responsible for several vital functions in mammalian cells, including adenosine triphosphate (ATP) production, steroid and stress hormones synthesis, cellular metabolism, and the induction of apoptosis [1–3]. Moreover, spatially, and functionally regulated specific microdomains known as the mitochondria-associated membranes (MAMs) between the endoplasmic reticulum (ER) and mitochondria are hot spots of Ca2+ transfer highlighting mitochondria vital role in cellular Ca2+ homeostasis [4]”. (lines 51-56)

3.2. Please replace “the molecular mechanism” with “possible molecular mechanism” since the precise mechanism was not described. 

ANSWER: Corrected to “possible molecular mechanism”

  1. METHODS

4.1. Please provide statement and number of the decision of the Ethical committee.

ANSWER: This is included in the Institutional Review Board Statement (page 9): The study was conducted according to the guidelines of the Declaration of Helsinki and approved by the Clinical Research Ethics Committee of Galicia (Reference code 2015/410). (lines 372-374)

4.2. Please provide intra- and inter- assay coefficients for all assays.

 ANSWER: Intra- and inter- assay coefficients were already calculated to get the graphical representation of the seahorse results (Figure 3). The basal oxygen consumption was measured 8 times followed by 4 measurement cycles after addition of the inhibitors, moreover, the experiment itself was repeated two times. Figure 3 represents the average oxygen consumption rate with the corresponding standard deviation. This information was added: “All data were obtained from two independent experiments, expressed as mean value ± deviation standard”. (lines 146-147)

  1. RESULTS

5.1. Please be more clear and precise in the description of the results.

ANSWER: We agree with the reviewer. The results section was updated and the following sentence was deleted “According to GERP, PhyloP, and phastCons, the splice site is well conserved. Pathogenicity was predicted using MutationTaster (prediction: disease-causing) and by calculating FATHMM and DANN scores, which corresponded to “damaging”.

5.2. Please replace “the molecular mechanism” with “possible molecular mechanism” since the precise mechanism was not described. 

 ANSWER: Corrected to “possible molecular mechanism”

(6) DISCUSSION

6.1. Please replace “the molecular mechanism” with “possible molecular mechanism” since the precise mechanism was not described. 

ANSWER: Corrected to “possible molecular mechanism”

6.2. Please discuss the association/correlation between different mutations and the manifestation of the symptoms in term of age and/or sex, as well as same/similar mutations in relation to the symptoms and age and sex.  

ANSWER: Following reviewer suggestion, discussion was improved and updated: The severity of Sengers syndrome can vary widely depending on the nature and location of the variant (Table S4, Supplementary Material) [9,17]. For instance, one of 2 siblings (P4 and P5) carrying the homozygous variant c.1131+5G>A, which leads to a splicing defect, died at age 12, while the other was still alive at age 10 [23]. Patient P3 with the same homozygous variant had a later onset of the disease and was still alive at the age of 41 [24]. On the other hand, isolated congenital cataracts was the only clinical sign observed in three patients (P13, P14 and P15) carrying the homozygous p.Ala142Thrfs*4 mutation, which resulted in complete deletion of exon 8 and premature truncation of the protein [22]. It seems homozygous variants lead frequently to a fatal prognosis of Sengers syndrome, while patients with compound heterozygous variants might live longer. Moreover, the age of onset at birth is probably associated with protein-truncating variants such as nonsense, frameshift, splice acceptor, and splice donor mutations. (lines 171-182)

  1. GENERAL. Please use official abbreviations.

ANSWER: Following reviewer suggestion, we have corrected the following abbreviations:  adenosine triphosphate (ATP)

Reviewer 2 Report

Barbarosa-Gouveia et al. submitted an original paper describing the new point mutation in the AGK gene associated with fatal Sengers syndrome. The article is generally well-written and contains novel data. To improve the quality of the presented data, I suggest a few minor corrections:

  1. Please, address in the manuscript the already known point mutations in the AGK gene correlated with Sengers syndrome (for example, a Table including the type of mutation and reference). Did you verify the presence of the already described SNPs ?
  2. Materials and Methods Section: all reagents should be followed by the producer, town, state, and country of their origin.
  3. Please, include the statistical analysis in the graphs (Figure 3)
  4. Would you please correct the Supplementary Materials file and remove the track-changes format? 

Author Response

Barbosa-Gouveia et al. submitted an original paper describing the new point mutation in the AGK gene associated with fatal Sengers syndrome. The article is generally well-written and contains novel data. To improve the quality of the presented data, I suggest a few minor corrections:

ANSWER: We would like to thank the reviewer for a careful and thorough reading of our manuscript and the very positive and kind comments.

  1. Please, address in the manuscript the already known point mutations in the AGK gene correlated with Sengers syndrome (for example, a Table including the type of mutation and reference). Did you verify the presence of the already described SNPs ?

ANSWER: We have included the information of the already known point mutations in AGK in the Supplementary Material, Table S4. The variant described in the manuscript is reported for the first time.

  1. Materials and Methods Section: all reagents should be followed by the producer, town, state, and country of their origin.

ANSWER: Following reviewer suggestion, we have updated all the reagents detailed information.

  1. Please, include the statistical analysis in the graphs (Figure 3)

ANSWER: Following reviewer suggestion, we have added the sentence “All data were obtained from two independent experiments, expressed as mean value ± deviation standard.” in the graph, figure 3. (lines 146-147)

  1. Would you please correct the Supplementary Materials file and remove the track-changes format?

ANSWER: We apologize for this. The Supplementary Material was updated and corrected.